# Significant Differences and Experimental Designs Do Not Necessarily Imply Clinical Relevance: Effect Sizes and Causality Claims in Antidepressant Treatments

**DOI:** 10.3390/jcm12093181

**Published:** 2023-04-28

**Authors:** Iván Sánchez-Iglesias, Celia Martín-Aguilar

**Affiliations:** 1Department of Psychobiology & Behavioral Sciences Methods, Complutense University of Madrid, 28223 Madrid, Spain; 2Centro Universitario San Rafael-Nebrija, Universidad Nebrija, 28036 Madrid, Spain; cmartina3@alumnos.nebrija.es

**Keywords:** antidepressants, experimental design, randomized clinical trial, RCT, significance testing, clinical relevance, effect size, causality

## Abstract

Clinical trials are the backbone of medical scientific research. However, this experimental strategy has some drawbacks. We focused on two issues: (a) The internal validity ensured by clinical trial procedures does not necessarily allow for generalization of efficacy results to causal claims about effectiveness in the population. (b) Statistical significance does not imply clinical or practical significance; *p*-values should be supplemented with effect size (ES) estimators and an interpretation of the magnitude of the effects found. We conducted a systematic review (from 2000 to 2020) on Scopus, PubMed, and four ProQuest databases, including PsycINFO. We searched for experimental studies with significant effects of pharmacological treatments on depressive symptoms, measured with a specific scale for depression. We assessed the claims of effectiveness, and reporting and interpreting of effect sizes in a small, unbiased sample of clinical trials (*n* = 10). Only 30% of the studies acknowledged that efficacy does not necessarily translate to effectiveness. Only 20% reported ES indices, and only 40% interpreted the magnitude of their findings. We encourage reflection on the applicability of results derived from clinical trials about the efficacy of antidepressant treatments, which often influence daily clinical decision-making. Comparing experimental results of antidepressants with supplementary observational studies can provide clinicians with greater flexibility in prescribing medication based on patient characteristics. Furthermore, the ES of a treatment should be considered, as treatments with a small effect may be worthwhile in certain circumstances, while treatments with a large effect may be justified despite additional costs or complications. Therefore, researchers are encouraged to report and interpret ES and explicitly discuss the suitability of their sample for the clinical population to which the antidepressant treatment will be applied.

## 1. Introduction

### 1.1. Background

Depression affects 3.8% of the population, about 280 million people worldwide. It is a common illness, and may be a serious health condition of moderate-to-severe intensity when recurrent [1]. In addition, in fact, most people with a history of either a major depressive episode or major depressive disorder have had recurrent episodes [2]. Unlike common mood fluctuations, depression is a disorder characterized by low mood and energy, decreased interest in activities considered pleasurable, and negative self-esteem [3]. More than 700,000 deaths by suicide per year are attributed to depression [1]. Depression is a public health problem with an increasing trend in prevalence [4].

Psychological treatments are effective in the treatment of depression [5]. However, the lion’s share of interventions are based on pharmacological treatment; prescription of antidepressants is increasing year on year, more than the increase in population and in the incidence of depression [6]. Many experimental studies, in the form of randomized clinical trials (RCT), have found a significant effect of a drug by comparing groups of subjects given the drug with a placebo control group. However, not all antidepressant treatments work in clinical practice. We are referring to a relevant number of failed interventions, beyond anecdotal individual cases [7,8]. Some studies have addressed treatment-resistant depression, a phenomenon with no consensus definition that hinders treating patients with this condition [7,9,10]. Some factors of a failed antidepressant treatment may be related to the stigma associated with the use of antidepressants [11] and patient adherence in general [7,10], and pharmacokinetic and/or pharmacodynamics factors [7], but also to the antidepressant medication of choice, dosage, duration of the treatment [12], and the definition of a successful intervention (remission of the symptoms in a clinical trial versus the clinical reality of the patients’ everyday life) [10]. In our view, this difference between the efficacy found in clinical trials and the effectiveness once the treatments are prescribed in the clinical setting may also be related to the clinical trials themselves. Clinical trials have been criticized on many grounds, from their design to the reporting of results [13], the existence of an industry bias in studies sponsored by manufacturing companies [14,15], or inadequate follow-ups [15]. In the field of depression, a meta-analysis conducted by Cipriani [16] has been criticized for failing to interpret the practical significance of its results and for its methodological biases [17]. Despite these flaws, the meta-analysis has been used by opinion leaders and the media to settle the debate on the effectiveness of antidepressants. Other criticisms of clinical trials on antidepressant treatments relate to ineffective double-blinding procedures [18], the use of response or remission rates based on continuous data [19], and the failure to account for poor outcomes in longitudinal follow-up studies [20].

In this paper, we want to focus on two specific aspects: (1) the misuse of clinical trials as a tool that allows us to draw conclusion about the population from a controlled design, and (2) the absence of calculation and interpretation of effect size (ES) estimators in published papers.

### 1.2. Experimental Design in Clinical Trials and Its Role in Internal and External Validity

Broadly speaking, the classification of scientific studies can be made based on their methodology, as follows: experimental, quasi-experimental and non-experimental studies. All three methodological approaches have their place in scientific research. However, experimental designs (randomized controlled studies) are the most frequent in the biomedical field, at least when testing methods of treating a disease or medical condition.

In experimental designs, the researchers actively manipulate one or more independent variables (the presumed causes) before observing variations in the dependent variable. At the same time, controlling for extraneous variables allows for the ruling out of alternative explanations of that variation. There is a well-established inferential superiority of randomized controlled experiments (sometimes simply called experiments, or RCT in the biomedical field) over the other two methodologies [21,22]. These studies have greater internal validity, and make it possible to establish a causal relationship between the study variables, although this inference is of a practical nature [22,23,24]. The design, analysis, and interpretation of the results of an RCT vary depending on the scientific hypothesis being tested [25]. Superiority trials, for example, assess the efficacy of an intervention against an active control (standard therapy) or placebo group, with the hypothesis that the intervention is superior. Non-inferiority and equivalence trials, on the other hand, compare a novel intervention with a standard therapy, with the hope that the intervention is either not inferior (and possibly better) than or equivalent (neither better nor worse) to the active control. These last two trial types may be designed for ethical or practical reasons. On the one hand, it would not be ethical to compare the novel treatment with a placebo group when an established and effective alternative is available. Moreover, equivalence and non-inferiority trials are quicker and cheaper. On the other hand, equivalence and non-inferiority trials are not suitable when there is no effective treatment and a placebo group must be used. Additionally, equivalence and non-inferiority trials can be challenging, and may encounter validity issues [26,27] and ethical concerns [28].

Regardless of the objective and design of an RCT, a significant challenge is that the patients and treatments used in clinical trials may not be representative of those used in clinical practice [12]. Clinical trials have strict patient selection criteria, in the form of inclusion and exclusion criteria, to enhance internal validity, but at the cost of compromised external validity, especially ecological validity. These criteria include factors such as gender or age range, certain scores on clinical scales, comorbidities, diseases or disorders, substance abuse, suicide risk, contraindication, hypersensitivity or intolerance to certain substances or drugs, non-response to medications, taking certain other drugs, or pregnancy or breastfeeding, among others. As a result, these criteria may exclude a significant portion of the population with depressive symptoms, making it impossible to prove the effectiveness of the treatment for those individuals. Therefore, generalization of results from RCT should be made with caution, and conclusions about the effectiveness of an antidepressant treatment should be made only in the context of the clinical trial itself. However, the fact that researchers may wrongly assume that high internal validity allows them to draw conclusions about the entire population of people with diagnosed depression is a serious problem. This issue has been a topic of discussion in medical research for some time [29], and remains a current topic [30].

### 1.3. Null Hypothesis Significance Testing, ES, and Clinical Significance

Null hypothesis significance testing (NHST) is the most commonly used method for inference in scientific research, but it has faced frequent criticism [31,32,33]. However, some of the criticisms may stem from the misuse of NHST by researchers and/or a lack of understanding among readers [34,35,36]. NHST involves the use of *p*-values to make categorical judgments about accepting or rejecting the null hypothesis, which may lead to oversimplified or wrong conclusions [17]. Furthermore, often the acceptance or rejection of the null hypothesis marks the conclusion of the analysis. Once researchers identify a significant result and establish the presence of an effect, they tend to overlook the extent or magnitude of that effect (i.e., the ES). The effects of two treatments are always different, if only by a small value, even to a few decimal places. Increasing the sample size will eventually yield a statistical significance [37], so the question of whether they are different is a trivial exercise [38].

ES does not replace NHST’s purpose, which is to determine statistical significance, but provides additional information on the size of the observed effect. It helps answer the question, “How large of an effect can be expected in the population?”. In addition, reporting an ES estimate is not enough; it has to be interpreted in the context of the study. To assess the practical relevance of a significant effect, authors can compare it with well-stated effects from similar research settings or everyday examples. If no benchmarks are available, arbitrary but published thresholds can be used as a reference. However, labels such as “small,” “medium,” or “large” may be misleading or uninformative. Researchers tend to use these labels to interpret ES indices such as Cohen’s d, but Cohen himself warned about the dangers of the abuse of arbitrary labels [39] (p. 13).

Recommendations for calculating, reporting and interpreting ESs, whenever possible, are present in the ASPA style guide [40], which is widely used in the Social Sciences, and other fields such as international businesses [41]. In the medical field, there are examples of journals recommending or even requiring the use and reporting of ES estimators [42,43,44,45,46,47]. Other journals do not [48,49], although some of them explicitly point out that *p*-values “provide no information about the size of an effect or an association” [49]. In any case, none of the aforementioned papers indicates that the ES estimators should be interpreted in a practical or clinical context. Some studies have found that NHST *p*-values were rarely supplemented with information on ES, despite the existence of published guidelines [50,51]. In the field of substance abuse, a systematic review [52] revealed that 57.9% of the selected studies did not report ES indices, while 47.4% did not interpret the magnitude or relevance of their findings. Similarly, a review examining the effects of physical exercise on depressive symptoms [53] discovered that 52.4% of experimental studies did not report ES indices. Of the remaining studies, 23.8% reported statistics but did not provide interpretation, while another 23.8% used Cohen’s arbitrary thresholds without contextual meaning.

In summary, authors are expected to systematically report and interpret ESs in studies using inferential statistics, and even more so in randomized, controlled experimental studies. However, based on our previous work in the clinical setting [52,53], we believe that we are not there yet.

### 1.4. Objective and Hypotheses

The goal of this study was twofold. By searching several databases, a systematic review was conducted to obtain an unbiased sample of experimental studies (i.e., clinical trials or randomized controlled studies) on the effectiveness of antidepressant treatments. The selected studies were analyzed to:(a)Examine causality claims about treatment efficacy not only in that particular trial with its specific clinical conditions, but effectiveness in the everyday life of the population affected by depression. From the literature, it is anticipated that some studies will not acknowledge the limited generalizability of their experimental findings to the population.(b)Determine the proportion of studies that used ES estimators, and the proportion of studies that interpreted the magnitude of statistical effects found (interpreting the reported ES estimators or otherwise). Our hypothesis is that, similar to other areas of scientific research, approximately 50% of studies examining antidepressant treatments will not report ES indices, and a similar proportion will fail to provide an interpretation of the magnitude of their significant results.

## 2. Methods

We followed the PRISMA guidelines for systematic reviews [54] to obtain an unbiased sample of studies. Below, we make explicit the steps taken by the researchers (Section 2.1, Section 2.2, Section 2.3, Section 2.4, Section 2.5 and Section 2.6).

### 2.1. Eligibility Criteria

In order to address the objectives, we needed the selected studies to use experimental methodology. Additionally, we only selected studies with significant results on depressive symptoms. This is because researchers are not expected to report effect sizes or make claims about effectiveness when statistically significant effects have not been found. These were necessary criteria, even though this drastically reduced the number of articles in the final sample.

To summarize the inclusion criteria, in order to be included in this review, the studies had to be published between 2000 and 2020 in a peer-reviewed scientific journal, in Spanish or English. Other inclusion criteria were: (a) conducting randomized controlled studies, (b) comparing the effect of pharmacological treatments (with each other and/or with a placebo group), (c) on depressive symptoms, (d) measured with a specific quantitative scale for depression, and (e) with at least one statistically significant result in the main outcome.

### 2.2. Information Sources

The authors conducted a systematic literature search for relevant studies. The following ProQuest databases were used, for the period from 2000 to 2020: PsycINFO, PsycARTICLES, Consumer Health Database, Psychology Database and Public health Data Base. The search was repeated in Scopus and PubMed.

### 2.3. Search Strategy

The same search queries were typed in each chosen data base, in both English and Spanish, using the following Boolean expression: (“clinical trial” OR “experiment”) AND (“antidepressant” OR “depression”) adapting the syntax to the specific rules of each database engine. The search was restricted by title. The eligible results were restricted to peer-reviewed papers published in scientific journals. Theses and dissertations, chapters, books, and gray literature items were excluded. The publication date was limited to the year range from 2000 to 2020, inclusive. In addition, the references of articles not selected in a first filter, such as reviews or meta-analyses, were reviewed to complete the search.

### 2.4. Selection Process

We imported the data from the earlier step into a single Excel file in order to find and eliminate duplicate records. Two reviewers separately assessed each record’s title and abstract to determine whether it was appropriate for retrieval and reading. Consensus was reached among the reviewers to resolve disagreements. Whenever necessary, a third researcher provided support in reaching the final decision.

### 2.5. Data Collection Process

Both authors, on their own, retrieved all eligible records, and read these reports to determine their final inclusion and, if included, extract the relevant data.

### 2.6. Data Items

On their own, each reviewer checked the methodology, and extracted statistical analysis techniques, ES estimators, and interpretations (if any) from each selected study. They also searched and excerpted claims about effectiveness of the treatment in the population beyond the context of the experiment.

Each reviewer independently sought and retrieved the methodology and statistical analysis techniques for each report chosen, as well as the ES estimators and ES interpretations, if applicable. The Results section of each study was searched for the ES estimators (contextual or statistical). Some examples of the effect size measures we could find in the selected studies belong to the *d* family, which includes Cohen’s *d* and Hedges’ *g*. A transformation of *d*, which can also be computed independently, is the correlation coefficient *r* or its square, *r*^2^. When it comes to ANOVA as a statistical analysis, we can expect η^2^ or a more unbiased ω^2^. For the analysis of categorical data, we could find the contingency coefficient, Cramer’s *V*, phi coefficient, or odds ratio. Other less common indices can also be found, but we have mentioned the most widely used ones [55,56].

The reviewers also searched the Results and Discussion sections of each report for interpretations of the statistically significant findings based on ESs. Studies were classified based on their methodology, main outcome data analyses, ES indices reported (whether explicitly as ES estimators or not), and the interpretation of the magnitude of the statistically significant effects found (whether explicitly reported as such or not). As in the previous step, disagreements were resolved by consensus and with the help of a third researcher, if needed.

Regarding the second objective, we examined studies in search of causal claims of treatment effectiveness in the population diagnosed with depression. We also recorded the proportion of studies that, without making such claims, do not explicitly state the limitation in generalizing their results to the population.

## 3. Results

### 3.1. Study Selection

We screened 1144 studies, after discarding 878 duplicates from an initial sample of 2028 records. Of those 1144 studies, we excluded 1130 by either title or abstract. The remaining fourteen were assessed for eligibility, and four were finally discarded because they did not report significant results for the relevant outcomes [57,58,59,60]. Therefore, 10 clinical trials were included in the review. Figure 1 displays the flow diagram of the search and selection of studies.

### 3.2. Characteristics of Studies and Patients

Among the selected studies, we identified eight superiority studies, one non-inferiority study, and one study that employed both superiority and non-inferiority analyses. Five out of ten studies were carried out in the USA [61,62,63,64,65], two in Iran [66,67], one in Korea [68], and one in France [69]. The duration of the treatments (and, if applicable, the follow-up) ranged from 6 to 24 weeks. The specific measures of treatment efficacy on depression found were the Montgomery–Asberg Depression Rating Scale (MADRS), Hamilton Depression rating scale (17 and 24 item versions) [70], and Beck Depression Inventory (BDI). As the main efficacy outcome, two out of ten studies used HAMD, four used MADRS, three used both, and one used BDI. Other characteristics of the studies, such as the number of groups, the treatment they received, and the statistical tests used, can be found in Table 1.

To be eligible for participation in the selected clinical trials, patients needed to be adults and had to meet all of the inclusion criteria and none of the exclusion criteria. These criteria were related to specific factors such as:Age range [61,62,63,64,65,66,68,69,71];Exceeding a certain weight [64];Exceeding a certain threshold score on one or more clinical scales [62,63,65,66,68,69,71];The presence of certain comorbid disorders [61,62,63,64,65,66,68,69,71];Lack of response to antidepressant therapy during current episode [61,62];Substance abuse [61,62,63,64,65,66,69,70,71];Suicide risk or ideation [61,62,63,64,65,66,69,71];Hypersensitivity or non-response to certain medicaments [63,64,65,71];Taking certain others [62,63,65,71];Intolerance to certain excipients such as lactose [71];Pregnancy or breastfeeding [63,64,66,67,68,69,71];Contraindication for certain drugs such as aspirin [67];Cardiovascular or neurological diseases [62,64,67,68];Developmental disorder [63];Mental retardation [63];Risk of seizure [63,66];Personality disorder [68];Risk of poor compliance [69];Recent hospitalization [68];Hospitalization due to severe depression [69];Other medical conditions [62,63,64,66];Electroconvulsive therapy [66,68];Receiving other treatments related to depression [63,66,68,69];Having previously participated in a clinical study [63,68].

### 3.3. Claims about Efficacy and Efectiveness

Out of ten studies, only three explicitly acknowledged the difficulty of generalizing the efficacy results to effectiveness in day-to-day clinical practice. These expressions were: “Generalizability of the study findings is limited by the small sample size and enrollment criteria that excluded individuals...” [61] (p. E8), “...the ability to generalize the results to typical outpatients is somewhat limited since patients were selected from among those with very limited to no comorbid medical conditions and few concomitant medications” [62] (p. 468), and “The generalizability of the results from a controlled clinical trial to real-world clinical practice can be a potential concern” [71] (p. 1613).

On the other hand, some expressions from other studies may mislead the readers into thinking that the efficacy results imply effectiveness in the population. Some examples are: “The results of this study provide statistically significant support for enhancement of the antidepressant effect of the SSRI fluoxetine by concurrent treatment with modafinil”, “… seems to confirm the early slight to medium clinical benefits of lithium combined with an antidepressant […] concerning unipolar patients with severe major depression…”, and “The efficacy of duloxetine at doses of 80 and 120 mg/day in the treatment of MDD was demonstrated”.

### 3.4. Effect Size Estimators and Their Interpretation

Out of ten studies, only two explicitly reported ES indices: one used Cohen’s *d*, while the other utilized a descriptive score change from baseline. Furey and Drevets [64] used Cohen’s *d*, the standardized mean difference, as an ES indicator. It indicates the number of standard deviations that separate the sample means. Wade et al. [71] used the mean change from baseline. Unlike Cohen’s d, it is the unstandardized difference between sample means. This provides less information about the magnitude of the difference, as it depends on the metric of the instrument.

Four studies interpreted the magnitude of their findings by considering contextual information. For instance, one study noted that “The size of the medication-placebo difference was […] larger than the mean difference from placebo seen at 6 to 8 weeks in antidepressant studies in the US Food and Drug Administration database” [61] (p. E5). Another study mentioned that “Estimated probabilities of remission […] are similar to those reported previously for patients receiving these doses of duloxetine” [62] (p. 466). Similarly, another study found that “In such studies, our results on effectiveness were comparable with those from other clinical trials as well as other studies without commercial pharmacogenomic testing” [68] (p. 477). On the other hand, five studies neither reported ES estimators nor interpreted the magnitude of their significant results. Only one study did both.

## 4. Discussion

In the present study, we have discussed the importance of two methodological issues for the interpretation of results from experimental studies. We have examined to what extent they are present in an unbiased sample of clinical trials on the effectiveness of antidepressant treatments published between 2000 and 2020. Regarding the first issue, the presence of statements about causal effects of treatment on a population very different from the clinical trial samples, we found that most of the selected studies failed to explicitly acknowledge that efficacy does not imply effectiveness. Regarding the second issue, the calculation, reporting, and interpretation of ES estimates, we found that, as expected, most of the studies failed to do it appropriately.

### 4.1. Balancing Internal and Ecological Validity in Clinical Trials: Implications for Practice

As previously discussed in the Introduction, our review has found that the characteristics of patients recruited for clinical trials may differ significantly from those encountered in everyday clinical practice. It is undeniable that inferences derived from experimental studies (and therefore clinical trials) have the best guarantees for establishing causal relationships. However, the same control of internal validity causes problems of ecological validity. Depression is a condition that is affected by multiple factors in addition to medication. Therefore, finding significant effects (even if these effects are large) of a medication in a trial is not sufficient to assume also effectiveness in those same patients in their daily life, especially in long-term treatment.

We agree with Zarin et al. [12] that the extent to which RCT results should be used to guide routine clinical decisions needs to be studied, as many patients are receiving treatments without empirical support for their specific characteristics. One possibility is to supplement the results of clinical trials with information received from non-experimental studies, with less internal validity but greater ecological validity of the clinical setting. It is not always necessary to conduct an experimental study to demonstrate an effect, and sometimes an experimental study can be so far removed from applied clinical reality that its good internal validity is of no use. Some studies have, tongue in cheek, effectively argued these ideas. One such study, in a systematic review, found that the efficacy of parachutes in preventing mortality in falls had not been proven in clinical trials [72]. Another study, in a randomized clinical trial, found that the non-parachute group did not have a higher mortality or injury rate (0%) than the parachute group (0%) when jumping 0.6 m from a stationary aircraft [73]. However obvious this may seem, the results of the present study suggest that we are far from making these issues routine.

In any case, one way to avoid conveying a misleading message to readers, particularly those who are not trained in research methods, is to routinely acknowledge in every clinical trial that the study sample may not be representative of the population seeking treatment in everyday clinical practice.

### 4.2. The Need for Encouraging ES Estimators in Scientific Reports

Even if we could draw conclusions about the effectiveness of a drug (versus another drug or versus no drug) on the population of people diagnosed with depression, the results of a clinical trial are of limited utility if the inferential statistics for NHST are not accompanied by ES estimators, as statistical significance does not imply clinical or practical significance. It will be difficult for health professionals to know whether an appreciable effect is expected, or whether it is so small that it can be easily masked (or counteracted) by other factors in the day-to-day lives of these patients. Focusing solely on *p*-values distracts the researchers from the practical significance or usefulness of a statistically significant effect [38]. The researchers who collected the data must interpret them and decide on the practicality of their findings, even if this introduces a certain amount of subjectivity. It is in the context in which the scientific hypothesis is set that the results of a study have to be interpreted. ES estimators help us to quantify the magnitude of the effects found, and should be reported whenever possible. However, it is the researcher, using theory and context, who has to interpret them.

A possible recommendation to avoid the absence of ES estimators (and their interpretation) in clinical trial reports would be for all journals to have specific instructions for authors. However, in 2017, Dexter and Shafer reviewed the biomedical literature and assessed the impact of guidelines, instructions to authors and statistical checklists [74]. They found that instructions and checklists are not sufficient to improve the statistical methods and reporting, and that journal editors and reviewers may lack the statistical expertise to enforce compliance with their own guidelines. Although instructions and checklists are a useful tool for authors to check they are not skipping an important step in reporting, we adhere to the recommendation to apply “a policy of 100% statistical review of all manuscripts that have any form of data” (p. 946), as guidelines are no substitute for statistical review [74].

### 4.3. Limitations

We assessed studies on the effect of antidepressant treatments, focusing on (a) the reporting and interpreting of ES and (b) the causality claims of the effect of such treatments on the population of patients diagnosed with depression. It is important to note that this study has certain limitations.

First, the search terms used were limited. Another, more comprehensive search equation may have been used, presumably obtaining a different list of records. Furthermore, it is worth noting that the search was restricted to titles only, which may have excluded other experimental studies that reported their study design in the abstract but not in the title. Additionally, the review was confined to Spanish and English languages, as the authors were proficient only in these languages. Consequently, there may be other studies published in different languages that could be relevant to our research question. Finally, we selected studies only from papers published in peer-reviewed scientific journals. Although these papers are generally considered the most reliable sources of information, other technical reports or dissertations could have added records to the initial list. As a consequence, our search yielded a relatively small number of studies (*n* = 10); we considered this a suitable sample, according to the study’s objectives and proposed inclusion and exclusion criteria. However, we acknowledge that the limited number of studies may impact the generalizability of our findings. We proceeded in this manner because our research questions were methodological in nature, and we did not intend to conduct an exhaustive search of the state of the art in experimental treatments and their effects on depression. To acquire a larger sample of publications, an option might have been to conduct the search using more databases, a wider range of publication dates, synonymous search phrases, etc. However, our goal was to discuss the importance of having estimates of the magnitude of the statistical effects found in clinical trials, and a proper approach to causality claims when generalizing the results of the trial to the population. This can be carried out by obtaining an unbiased, albeit arbitrary, selection of articles through a systematic review process. We do not suspect that our sample, despite being small, differs from that of other studies that may have been left out of our selection. Nonetheless, given these limitations, we should generalize our results with caution. Future studies may use other search terms, publication dates, databases, etc., to address this objective.

### 4.4. Future Research and Implications for Practice

Future research could address the same issues not only in biomedical fields but also in the social sciences, which are susceptible to suffering from them. For instance, in the treatment of depressive symptoms, research could explore psychological or behavioral interventions. At universities, we strive to educate future professionals and researchers about the importance of methodological aspects in conducting studies. This includes interpreting both the external validity of a study, which leads to causal claims about the population, and the magnitude of the findings. However, we have found that scientific publications often overlook some good methodological practices. Future publications could emphasize these aspects, with the intention of improving the writing of papers, facilitating comprehension for readers, and ultimately benefiting both the scientific and clinical communities.

It seems appropriate to compare the experimental results on the effectiveness of an antidepressant drug with those from observational studies. This would allow clinicians to have greater flexibility in prescribing medication to their patients, based on their clinical and sociodemographic characteristics. The first step is, as we have mentioned throughout this work, to always remember that clinical trial participants have very specific characteristics, to maintain internal study control.

Furthermore, knowing the size of the effect of a treatment in its context has implications for decision-making. Treatments with a small effect may be worthwhile if their application is easy, cheap, and has fewer adverse effects, or for any other practical reason. However, if none of these circumstances apply, switching from a standard treatment to a novel one (or incorporating a treatment for people who do not have it) in exchange for a slight improvement may not be convenient. On the contrary, treatments with a large effect (in our example on depressive symptoms) may be worthwhile, even if they are more expensive or have some complications in their prescription.

We wholeheartedly believe that an improvement in these two aspects, by researchers publishing in the field of antidepressant treatments, will result in better-informed clinical practice and decisions that are better tailored to patients suffering from depression.

## 5. Conclusions

Results about the efficacy of antidepressant treatments, derived from clinical trials, often influence daily clinical decision-making. This study aims to encourage reflection on the applicability of these results. Working with patients diagnosed with depression, clinicians are urged to reflect on the similarities between their patients and the participants in clinical trials. This reflection may lead them to search for observational studies that can complement their knowledge and help them make informed decisions. For researchers who conduct and publish clinical trials, we encourage reflection on the importance of reporting, along with the rest of the results, and on the magnitude of the effects found, and encourage them to provide an explicit discussion of the similarity of their sample to the clinical population to which the antidepressant treatment will be applied.

## Figures and Tables

**Figure 1 jcm-12-03181-f001:**
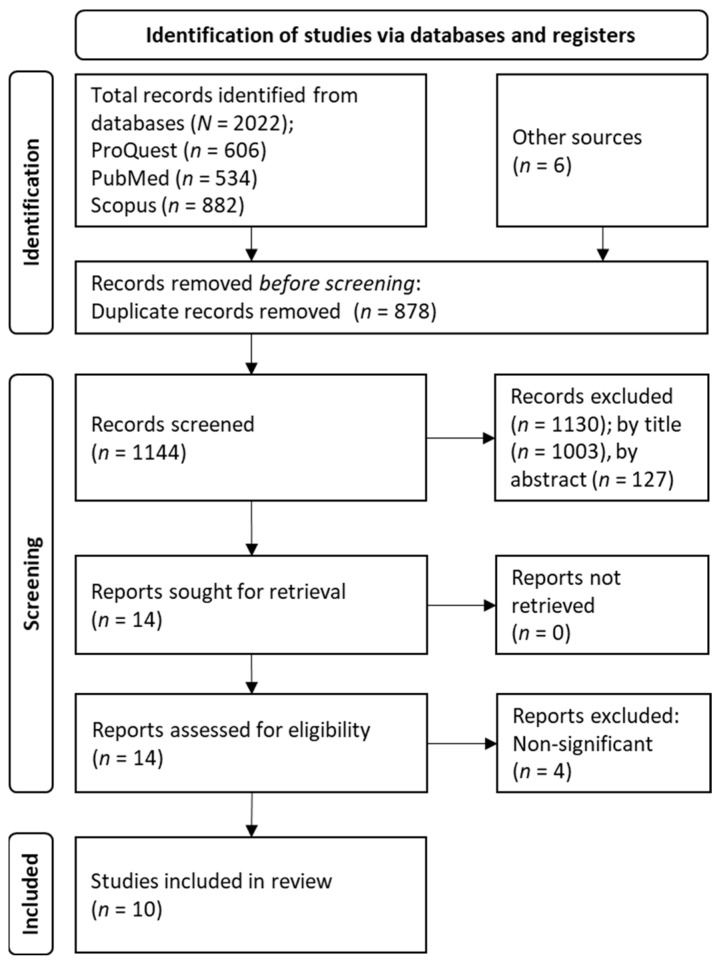
Flow diagram for the systematic review.

**Table 1 jcm-12-03181-t001:** Design, Main Statistical Analyses and Effect Size (ES) Estimates and Interpretations in the Studies Selected.

Citation.	Efficacy Measure	Groups	Duration (Weeks)	Scale	Statistical Test	ES	ES Inter.
Abolfazli et al., 2011 [66] (Iran)	Total/Response/Remission	Fluoxetine-modafinil vs. Fluoxetina-placebo	6	HAMD-17	*t*-test/ANOVA	No	-
Daly et al., 2017 [61] (USA)	Total/Response/Remission	Esketamine (28 mg) vs.Esketamine (56 mg) vs. Esketamine (84 mg) vs. placebo	76	MADRS	ANCOVA	No	Context
Detke et al., 2004 [62] (USA)	Total/Response/Remission	Duloxetine (80 mg) vs. Duloxetine (120 mg) vs. Paroxetine vs. Placebo	9 + 24	HAMD-17/MADRS	ANCOVA/MMRM/FET	No	Context
Han et al., 2018 [68] (Korea)	Total/Response/Remission	Neuropharmagen PGAT vs. treatment as usual	8	HAMD-17	ANCOVA/FET	No	Context
Januel et al., 2003 [69] (France)	% Pre-post reduction/Remission/Response	Clomipramine-lithium vs. Clomipramine-placebo	6	MADRS	*t*-test/ANOVA/FET	No	-
Khan et al., 2007 [63] (USA)	Total/Response/Remission	Escitalopram vs. Duloxetine	8	MADRS/HAMD-24	ANCOVA/MMRM	No	-
Furey and Drevets, 2006 [64] (USA)	Total/Response/Remission	Placebo vs. Scopolamine	4	MADRS	Mixed ANOVA	*d*	Context
Perahia et al., 2006 [65] (USA)	Total/Response/Remission	Placebo vs. Duloxetine (80 mg) vs. Duloxetine (120 mg) vs. Paroxetine	8 + 24	HAMD-17/MADRS	ANCOVA/MMRM	No	-
Sepehrmanesh et al., 2017 [67] (Iran)	Total	Sertraline-Aspirin vs. Sertraline-placebo	8	BDI	*t*-test	No	-
Wade et al., 2007 [71] (UK)	Total/Response/Remission	Escitalopram vs. Duloxetine	24	MADRS	ANCOVA	Mean change	No

Notes. ES: Effect size. ES inter: Effect size interpretation. Total: Pre-post scores difference. Remission: Remission rate. Response: Response Rate. MMRM: Mixed-Effect Model Repeated Measure. MADRS: Montgomery–Asberg Depression Rating Scale. HAMD-17: Hamilton Depression Rating Scale (17-item). HAMD-24: Hamilton Depression Rating Scale (24-item). BDI: Beck Depression Inventory. FET: Fisher’s exact test.

## Data Availability

Not applicable.

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
