# Peer review of "Significant Differences and Experimental Designs Do Not Necessarily Imply Clinical Relevance: Effect Sizes and Causality Claims in Antidepressant Treatments"

_jcm, 2023, doi:10.3390/jcm12093181_

Round 1
Reviewer 1 Report
The paper discusses two methodological issues in clinical trials on the effectiveness of antidepressant treatments. The first issue is the internal validity of clinical trial procedures, which may not allow the generalization of efficacy results to causal claims about effectiveness in the population. The second issue is the need to supplement statistical significance with effect size (ES) estimators and an interpretation of the magnitude of the effects found. The authors conducted a systematic review of a small but unbiased sample of clinical trials on the impact of pharmacological treatments on depressive symptoms measured with a specific scale for depression, published between 2000 and 2020. The review found that only a minority of the studies acknowledged the limitations of generalizing efficacy results to effectiveness claims, and few studies reported and interpreted ES indices. The paper concludes by encouraging clinicians to consider the applicability of results derived from clinical trials, seek supplementary knowledge from observational studies, and by urging researchers to report and interpret ES and discuss the similarity of their sample to the clinical population on which the antidepressant treatment will be applied.
One weakness of the abstract is that it does not provide more specific details on the systematic review methodology, such as the databases used, search terms, and inclusion/exclusion criteria. Additionally, the sample size of the included studies is relatively small, which limits the generalizability of the findings. The abstract could also benefit from more specific examples or implications of the issues discussed in the study.
The draft has no significant weaknesses, as it provides a clear and concise overview of the study's background, objectives, methodology, results, and discussion. However, providing more information on the inclusion and exclusion criteria used in the systematic review and the specific ES estimators used in the studies analyzed may be helpful. Additionally, the study could benefit from a more detailed discussion of the implications of the findings and suggestions for future research.
The methods section appears well-written and comprehensive regarding the steps taken to conduct the systematic review. However, some potential weaknesses could be addressed.
Firstly, the eligibility criteria only include studies published in peer-reviewed scientific journals in Spanish or English. This may have limited the scope of the review and excluded potentially relevant studies published in other languages or non-peer-reviewed sources.
Secondly, the search strategy only used a single Boolean expression and was limited to title searches. This may have missed some relevant studies that used different terminology or had relevant information only in the abstract or full text.
Finally, the review appears to have only included randomized controlled trials, which may limit the generalizability of the results to other study designs. Additionally, excluding studies that did not report significant results may introduce publication bias and exclude potentially valuable information.
Overall, while the methods section provides a clear and detailed description of the review process, there may be some areas for improvement regarding scope and search strategy to ensure a more comprehensive and unbiased review.
Reviewer 2 Report
This is a meta-analysis of clinical trials of antidepressants. The methodology of the study has major flaws. First, many antidepressant studies are not superiority studies, but rather ones designed to check that novel therapeutics are equivalent in efficacy. This aspect is not discussed. The result "we screened 1144 studies, after discarding 878 duplicates from an initial sample of 2028 records. Of those 1144 studies, we excluded 1130 by either title or abstract" is problematic, and many more than 14 studies were published in the period of 20 years, so the criteria with which the study titles were filtered need to be exactly specified, otherwise the results are not understandable.
In terms of actual reporting of results, the authors are seemingly confused with regard to two questions in clinical trials. One being question of generalizability/transportability, which is whether the trial results are applicable in real world settings. Two being the effect size reporting, meaning whether effect sizes need to be reported in a certain context.
I am not sure that the article is a research article, as it does not contain enough information in terms of novel findings. I suggest revise to a commentary/perspectives article.
Round 2
Reviewer 2 Report
I have no further comments.